# Analysis of the Coping Strategies of Primary Health Care Professionals: Cross-Sectional Study in a Large Brazilian Municipality

**DOI:** 10.3390/ijerph19063332

**Published:** 2022-03-11

**Authors:** Luciano Garcia Lourenção, Bruno Martinez Rigino, Natalia Sperli Geraldes Marin dos Santos Sasaki, Maria Jaqueline Coelho Pinto, Francisco Rosemiro Guimarães Ximenes Neto, Flávio Adriano Borges, Maria de Lourdes Sperli Geraldes Santos, José Gustavo Monteiro Penha, Daniela Menezes Galvão, Betânia Maria Pereira dos Santos, Isabel Cristina Kowal Olm Cunha, Jacqueline Flores de Oliveira, Max dos Santos Afonso, Carlos Leonardo Figueiredo Cunha, Francielle Garcia da Silva, Neyson Pinheiro Freire, Vagner Ferreira do Nascimento, Sidiane Teixeira Rodrigues, Taisa Moitinho de Carvalho, Messias Lemos, Juliana Lima da Cunha, Neide Aparecida Micelli Domingos

**Affiliations:** 1Nursing School, Federal University of Rio Grande, Rio Grande 96200-400, RS, Brazil; gustavo_penha02@hotmail.com (J.G.M.P.); dani.mgalvao@hotmail.com (D.M.G.); jacqueoliveira.enf@hotmail.com (J.F.d.O.); max.afonso@hotmail.com (M.d.S.A.); fran.garciasena@gmail.com (F.G.d.S.); sidiane.enf@hotmail.com (S.T.R.); juliana_cunhaa@outlook.com (J.L.d.C.); 2School of Medicine of São José do Rio Preto, São José do Rio Preto 15090-090, SP, Brazil; bmrigino@gmail.com (B.M.R.); nsperli@gmail.com (N.S.G.M.d.S.S.); psijaqueline@famerp.br (M.J.C.P.); mlsperli@gmail.com (M.d.L.S.G.S.); taisa.carvalho84@gmail.com (T.M.d.C.); micellidomingos@famerp.br (N.A.M.D.); 3Health Sciences Center, Vale do Acaraú State University, Sobral 62042-280, CE, Brazil; rosemironeto@gmail.com; 4Nursing Department, Federal University of São Carlos, São Carlos 13565-905, SP, Brazil; flavioborges@ufscar.br; 5Technical School of Health, Federal University of Paraíba, João Pessoa 58050-585, PB, Brazil; betaniamps@hotmail.com; 6Federal Council of Nursing, Brasília 70736-550, DF, Brazil; neysonfreire@gmail.com; 7Paulista School of Nursing, Federal University of Sao Paulo, São Paulo 04024-002, SP, Brazil; isabelcunha@unifesp.br; 8Nursing School, Federal University of Pará, Belém 66075-110, PA, Brazil; leocunhama@gmail.com; 9Nursing Department, Mato Grosso State University, Tangará da Serra 78300-000, MT, Brazil; vagnerschon@hotmail.com; 10Nursing Department, Federal University of Santa Catarina, Florianópolis 88040-370, SC, Brazil; messelemos01@gmail.com

**Keywords:** primary health care, family health strategy, health personnel, adaptation, psychological

## Abstract

Objective: To analyze the coping strategies used by primary healthcare (PHC) professionals. Methods: A cross-sectional, descriptive-analytical study realized with professionals working in primary healthcare units in São José do Rio Preto, a large city in the interior of São Paulo, Brazil. For data collection, we used an instrument developed by the researchers containing sociodemographic and professional variables, as well as the Problem Coping Modes Scale (EMEP). Results: We evaluated 333 PHC professionals. A difference was observed between the scores of the four coping strategies (*p* < 0.001), with the highest score for the problem-focused strategy (3.8) and the lowest score for the emotion-focused strategy (2.4). Physicians had the lowest scores in coping strategies focused on religious practices/fantastical thinking (*p* < 0.001) and pursuit of social support (*p* = 0.045), while community health agents had the highest scores in these coping strategies. Conclusions: Professionals working in PHC have different coping strategies for the problems and stressful situations experienced in the work environment. These strategies can involve more positive attitudes focused on confrontation and problem solving, and on emotional responses that involve attitudes of avoidance and denial of the problem.

## 1. Introduction

In Brazil, the establishment of the Unified Health System (UHS) after the promulgation of the Brazilian Constitution in 1988 determined the universal right of the population to health services. The system is organized by levels of complexity and Primary Health Care (PHC), or Basic Care (BHC), is the first level of health care [1].

PHC is considered the “entrance door” for users of the health system and is characterized by a set of health actions at the individual and collective levels that cover health promotion and protection, disease prevention, diagnosis, treatment, rehabilitation, harm reduction, health maintenance, and palliative care. At this level of health care, professionals develop comprehensive care actions that impact people’s health status and autonomy and the health determinants and conditions of the communities. The Family Health Strategy (FHS) is the priority model of health care in PHC [2,3].

Professionals in FHS teams work with client assignment, home visits, integrality of practice, and health promotion. They are multiprofessional teams, composed of a general physician (preferably a specialist in Family and Community Medicine), a nurse (preferably a specialist in Family Health), a nursing assistant and/or a nursing technician, and community health workers (CHWs) [2,4]. 

The organization and work process of PHC professionals can induce stress, which is understood as a psychological, physiological, and behavioral response of the worker to environmental, physical, and social situations that generate pressures and exhaustion [5]. The overload and the precariousness of the work conditions, the fragility of the employment bonds, the lack of training and perspectives of professional growth, the pressure for the fulfillment of goals, and the lack of professional autonomy are identified as the main causes of psychological suffering among PHC professionals [6,7]. 

To adapt to these stressful events, the workers expend a great amount of adaptive energy and, for this reason, in high levels or chronic situations, stress can cause physical and psychological illness, leading to difficulties facing problems and situations that feel threatening and/or wearying, and compromising the quality of life of workers [8,9].

When under stress, professionals seek coping strategies, which consist of cognitive, emotional, and behavioral reactions intentionally adopted to help them adapt to stressful situations and decrease susceptibility and negative impacts [10,11].

These coping strategies can be problem-focused (focused on ways to manage or solve the problem, re-evaluation, and positive meaning of the stressor), emotion-focused (involving avoidance behaviors, denial, emotions expressing anger and tension, attribution of blame, withdrawal from the problem, and ameliorative actions in the face of the stressor), based on religious or spiritually-related practices and fantastical thinking, or based on the pursuit of social support (pursuit of emotional or instrumental social support and the search for information) [12]. These coping strategies define different ways in which PHC professionals might deal with daily stress situations, avoid damage to health, produce motivation, and improve job satisfaction [13]. 

Moreover, coping strategies have an important protective role against the development of Burnout Syndrome (a state in which a person feels overwhelmed and immobilized, and feels reduced motivation) and directly influence the quality of life of professionals [14,15]. Coping strategies are used to reduce or eliminate suffering caused by stressors, and utilize internal and external resources such as social skills, beliefs, health condition, and availability of material resources [16,17].

Studies with health professionals in hospital services indicate that the most commonly adopted coping strategies are problem solving and social support, which consist of behavioral strategies aimed at solving the problem and cognitive strategies aimed at perceiving the stressor as a positive experience, in addition to seeking out information and socioemotional support [18,19]. 

A study with professionals from Australian emergency teams showed a higher adoption of positive coping strategies, such as social support and problem solving. These strategies have positive effects on the perception and management of stress, have a positive impact on work performance, and may decrease the risk of Burnout. For the authors, the implementation of resilience training programs by healthcare service managers can benefit professionals in coping with stress [20]. 

In Brunei, Asia, a study on nurses’ coping strategies highlighted that professionals who employ problem-oriented coping strategies have a lower risk of developing mental health disorders. On the other hand, those who adopt the emotion-focused strategies have better personality traits and are able to maintain a state of emotional arousal in the face of stressful events [21].

In Poland, nurses working in the care of patients with COVID-19 have mainly adopted the problem-focused strategies (actions that improve the situation) and the pursuit of social support (help and advice from others) for coping with the stressful situations caused by the pandemic [22].

In Brazil, a study with community health workers evidenced the adoption of positive coping strategies, such as problem-oriented ones and those based on the search for religious practices and fantasy thinking [23]. Similarly, a study with psychologists from a Psychosocial Care Center (CAPS) evidenced that problem-solving strategies are more used and better perceived by professionals [24].

In this context, we understand that PHC professionals may adopt different coping strategies when facing stressful situations in the work environment. In this study, we intend to explore which coping strategies PHC professionals use to adapt to stress. We understand that knowing the different coping strategies adopted by these professionals can contribute to the implementation of policies that promote positive coping mechanisms and improve the quality of life and productivity of this population, and, by realizing this, we can improve the mental health of PHC workers.

Thus, this study aims to analyze the coping strategies used by primary healthcare workers.

## 2. Materials and Methods

### 2.1. Type of Study

This cross-sectional, descriptive-analytical study was conducted in 2017 with professionals working in the primary healthcare units of a large municipality in the northwestern region of the state of São Paulo, Brazil. This municipality is the headquarters of the Regional Health Department (DRS) XV, which serves a total of 102 municipalities.

This large municipality had, at the time of the study, an estimated population of 451,354 inhabitants [25] and was administratively organized into five Health Districts. Primary Care comprised 27 health units, which had 58 teams of the FHS and 351 workers, they being 40 physicians, 78 nurses, 72 nursing assistants/technicians, and 161 community health workers [26].

### 2.2. Sample and Participants

All professionals of the FHS teams belonging to the 27 PHC units in the municipality were invited to participate in the study. The final sample was defined by convenience and included 333 professionals who responded to the data collection instruments, they being 32 physicians, 20 nurses, 71 nursing assistants/technicians, and 210 community health workers.

### 2.3. Procedures, Measurements, Variables and Outcome

For data collection, two self-administered instruments were used: first, a questionnaire developed by the researchers containing sociodemographic variables (gender, age, marital status, education, and family income) and professional variables (professional category, type of contract, weekly workload, time working in PHC, and satisfaction with the profession); and second, the Problem Coping Modes Scale (EMEP) developed by Seidl, Tróccoli, and Zannon [12] from the version adapted to Portuguese [27].

Factorial analysis of the EMEP scale indicates the usefulness of the instrument in research and intervention contexts, regarding coping with stress by different clientele [12,27].

The EMEP scale consists of 45 items, which are grouped to evaluate four factors: factor 1—problem-focused coping strategies; factor 2—emotion-focused coping strategies; factor 3—pursuit of religious practices/fantastical thinking; and factor 4—pursuit of social support. The items of EMEP scale encompasse thoughts and actions that individuals use to deal with the internal or external demands of a specific stressful event [12]. 

The answers to the items correspond to a Likert-type scale with five possible responses: 1 = I never do this; 2 = I hardly ever do this; 3 = I do this sometimes; 4 = I do this a lot; and 5 = I always do this. The characterization of coping occurs from the comparison between the mean obtained in each factor. The scores are obtained by the arithmetic mean, and the higher these averages, the higher is the frequency of use of the coping strategy [12].

Factor 1 [Coping focused on the problem] consisted of the following items: 1. I take the positive side of things into account; 3. I focus on some good that can come from this situation; 10. I insist and fight for what I want; 14. I find different solutions to my problem; 15. I try to be a stronger, more optimistic person; 16. I try to keep my feelings from getting in the way of other things in my life; 17. I focus on the good things in my life; 19. I accept someone’s sympathy and understanding; 24. I know what should be done and I am increasing my efforts to succeed; 28. I am changing and becoming a more experienced person; 30. I keep reminding myself that things could be worse; 32. I try not to act so hastily or follow my first idea; 33. I change something so that things will work out in the end; 36. I approach the situation in stages, doing one thing at a time; 39. I will come out of this experience better than I went into it; 40. I tell myself how much I have accomplished; 42. I have made a plan of action to solve my problem and I am sticking to it; and 45. I try not to close doors behind me. I try to leave open several ways out of the problem [12]. 

Factor 2 [Coping focused on emotion] consisted of the following items: 2. I blame myself; 5. I look for a culprit for the situation; 11. I refuse to believe this is happening; 12. I fight with myself; I keep talking to myself about what I should do; 13. I take it out on other people; 18. I wish I could change the way I feel; 20. I show anger toward the people who caused the problem; 22. I realize that I brought the problem upon myself; 23. I feel bad that I couldn’t avoid the problem; 25. I think people have been unfair to me; 29. I blame others; 34. I try to stay away from people in general; 35. I imagine and have wishes about how things could happen; 37. I find out who else is or was responsible; and 38. I think of fantastic or unrealistic things (like a perfect revenge or finding lots of money) that make me feel better [12].

Factor 3 [Coping based on religious practices and fantastical thinking] consisted of the following items: 6. I hope a miracle happens; 8. I pray; 21. I practice religion more since I have this problem; 26. I dream or imagine a better time than the one I’m in; 27. I try to forget the whole problem; 41. I wish I could change what happened to me; and 44. I cling to my faith to overcome this situation [12].

Factor 4 [Coping based on social support] consisted of the following items: 4. I try to keep my feelings to myself; 7. I ask for advice from a relative or friend whom I respect; 9. I talk to someone about how I’m feeling; 31. I talk to someone who can do something about my problem; and 43. I talk to someone to get information about the situation [12].

### 2.4. Statistical Analysis

The data were analyzed using the Statistical Package for Social Sciences (SPSS), version 20.0. To evaluate the coping modes, the mean scores of the four strategies (problem-focused, emotion-focused, religiosity/spirituality, and social support), the standard deviation, the 95% confidence interval (95% CI), and Cronbach’s alpha coefficient (α) were obtained. The internal consistency of the EMEP scale factors, measured by Cronbach’s alpha coefficient, was 0.879 for the problem-focused strategy, 0.804 for the emotion-focused strategy, 0.587 for the strategy based on religious practices/fantastical thinking, and 0.682 for the strategy based on social support. 

The comparison of coping strategies with the socio-demographic and professional variables of PHC workers was performed with the *t*-test for two means or analysis of variance (ANOVA) for three or more means, considering a significance level of 5% (*p*-value ≤ 0.05).

### 2.5. Ethical Considerations

Ethical approval regarding this study was obtained from the institutional ethics committee (decision: 1,776,737—16 October 2016; CAAE: 59604116.0.0000.5415). All the participants in this study were only included after informed consent had been obtained from them. All procedures performed in this study were in accordance with the ethical standards of the institutional research committee and with the comparable ethical standards outlined in the Declaration of Helsinki.

## 3. Results

A total of 333 PHC workers participated in this study, of which 32 (9.6%) were physicians, 20 (6%) were nurses, 71 (21.3%) were nursing assistants/technicians, and 210 (63.1%) were community health agents. Participants were predominantly female (81.1%), 60 years old or above (44.7%), with higher education (55.3%), married or in a stable union (63.1%), permanent employees (74.5%), with 40 working hours per week (91.0%), with family income of 2 to 10 minimum wages (56.8%), up to two years working in PHC (37.8%), and satisfied with the profession (77.8%) (Table 1).

As shown in Table 2, there was a statistically significant difference between the scores of the four coping strategies (*p*-value < 0.001). The highest score (3.8) corresponds to the problem-focused factor, which corresponds to behavioral strategies aimed at managing or solving the problem and cognitive strategies aimed at re-evaluation and positive meaning-making regarding the stressor. The lowest score (2.4) corresponds to the emotion-focused factor, that is, the cognitive and behavioral strategies that involve avoidance, denial, expressing emotions of anger and tension, attribution of blame, withdrawal from the problem, and ameliorative actions in the face of the stressor.

Analysis of the coping strategies according to the sociodemographic and professional variables of the PHC workers showed that there was no statistically significant difference (*p*-values > 0.05) between the scores of coping strategies and the type of contract, weekly work hours, and marital status. Therefore, we have not included these data in Table 3.

As shown in Table 3, physicians presented lower scores than community health agents in the coping strategies focused on religious practices/fantastical thinking (2.3 vs. 4.2; *p* > 0.001) and pursuit of social support (3.0 vs. 3.8; *p* = 0.045). Similarly, females showed higher scores than males for the coping strategies focused on religious practices/fantastical thinking (3.3 vs. 2.9; *p* = 0.001) and pursuit of social support (3.2 vs. 2.8; *p* = 0.005).

Younger workers (18 to 28 years old) had significantly higher scores than workers aged 60 years and older for the emotion-focused coping strategy (3.8 vs. 2.8; *p* = 0.014). Workers whose family income was up to one minimum wage presented higher scores for the coping strategy focused on religious practices/fantastical thinking than those with family income higher than 10 minimum wages (3.5 vs. 2.3; *p* = 0.001).

Workers with more than 10 years of professional PHC experience presented the lowest scores for the coping strategies focused on emotion (2.0; *p* = 0.008) and on religious practice/fantastical thinking (2.0; *p* > 0.001) than other workers. Furthermore, workers who reported being dissatisfied with their profession had higher scores than those who reported being satisfied with their profession for the coping strategies focused on emotion (2.6 vs. 2.4; *p* = 0.011) and religious practices/fantastical thinking (3.4 vs. 3.2; *p* < 0.010) (Table 3).

## 4. Discussion

The characteristics of PHC workers in this study are consistent with those described in other studies [28,29]. The predominance of female professionals is due to the feminization process that health professions have undergone in many countries, including Brazil, in recent decades [30,31]. Moreover, the professional structure of the teams is consistent with that proposed by the National Primary Care Policy, particularly with regards to the composition of the minimum number of FHS teams [2].

The coping strategy presented most among the PHC professionals in this study was the problem-focused one, which involves becoming aware of/identifying the stressor agent in order to manage or solve the problematic situation that causes them exhaustion/fatigue and stress. Through the process of reframing the problem, the professionals who adopt this strategy make cognitive efforts to perceive the problem in a positive way and, from there, attempt to face it [12,32]. These results corroborate the findings of other studies conducted with healthcare professionals that also presented the problem-focused strategy as the main coping strategy [23,33]. 

For the professional to develop effective coping skills, it is necessary to be aware of the presence of the stressor agent, which will allow him to adopt assertive coping mechanisms and achieve a reduction of occupational stress as a consequence [13]. In this sense, it is noteworthy that the way of coping depends on behavioral and cognitive strategies used to control external and internal demands, which generate overload and compromise the physical and mental capacity of the individual. Therefore, the coping process will depend on the interaction of the individual with the environment, and may be influenced by personality and previous experiences that allowed development of cognitive, behavioral, emotional, and social resources to face stressful situations [34,35].

It is notable that there is an important relationship between the coping strategies adopted and gender, as demonstrated in this study, whose results highlight that females showed higher scores than males for the coping strategies focused on religious practices/fantastical thinking and pursuit of social support. In addition to the fact that we have not identified studies highlighting the adoption of strategies focused on religious practice/fantasy thinking by men, the adoption of this strategy by the Brazilian population was reported as a characteristic of the female gender, regardless of the stressor agent [12,36].

We believe that the greater adoption of strategies focused on religious practice/fantasy thinking and seeking social support by female professionals may be related to cultural factors, such as the gender role division. While culturally, the man has the concrete responsibility of being the family provider, the woman inserted in the labor market is not exempt from her responsibilities of assisting family members [37]. Consequently, the way female professionals act can vary considerably according to the environment in which they find themselves (work environment vs. family environment). Therefore, they can entail difficulties in using problem-focused and emotion-focused strategies.

In addition, religiosity constitutes an effective resource for avoiding socially non-normative behaviors. We understand that the adoption of strategies focused on religious practice/fantasy thinking enhances women’s ability to establish self-control and self-regulation in emotional, cognitive (through beliefs), and behavioral aspects to achieve success in both work and family environments [38].

Similarly, the highest scores for the strategies focused on religious practices/fantastical thinking were presented among community health agents, female workers, those with family income up to one minimum wage, with more than 10 years of professional experience, and who reported being dissatisfied with their profession and whose dissatisfaction with their profession was identified as an important motivational factor. Contrary to what many people believe, turning to religiosity is a strategy that provides emotional support and favors a positive re-evaluation of a stressful situation [39].

A religious outlook in the workplace encourages the worker’s involvement with colleagues and favors the alignment of personal goals with organizational ones. Furthermore, spirituality is important for the construction of individual and community resilience, since it favors mutual support and enables groups (in this case, the FHS team) to share their needs, find strategies, and develop resources to face problems [40,41].

The strategy based on the pursuit of social support, more evident among community health agents and female workers, shows the importance of expanding the network of social and emotional support to PHC workers. Thus, it is essential that managers encourage team meetings, which are important opportunities for listening to the demands of these professionals and sharing information that contributes to more effective confrontation of the problems present in the work practice, thereby reducing the risk of suffering and mental illness of these workers [23].

In the case of females, facing the problems can become even more difficult, since female workers’ tasks are not over at the end of the workday, but can be but extend to domestic work and child care. Therefore, female workers are more susceptible to overload and emotional stress, and require social support to face the workplace problems [42].

The results also revealed that younger workers, those with more than 10 years of PHC experience, and those who are dissatisfied with their profession tend to adopt emotion-focused coping strategies; that is, they are more likely to develop denial and escape strategies, demonstrate feelings of tension and anger, and withdraw from problems and stressful situations. In this circumstance, it is common for workers to engage in behaviors that alleviate suffering, such as alcohol consumption and the use of illicit drugs [43,44]. 

Moreover, these professionals experience feelings of discouragement and hopelessness that compromise productivity and quality of life, and furthermore, hinder the adoption of more positive and assertive responses to a stressful situation [45,46]. Therefore, because these workers show difficulty in facing their problems, they deserve special attention from managers and other professionals within the team.

Finally, it is noteworthy that workers with more than 10 years of professional experience tend not to adopt emotion-focused strategies and/or religious practices/fantastical thinking. This may be due to their extensive professional experience, which brings the maturity needed to react in a more assertive, concrete, and objective way when faced with stressful situations, without great emotional attachment and free of unpleasant feelings [47,48].

The cross-sectional design of this study does not make it possible to establish cause-and-effect relationships, or perform further analysis on coping strategies and the sociodemographic and professional characteristics of PHC workers. However, the results should be carefully analyzed, since coping strategies are essential to solve problems and manage stressful situations that permeate the PHC work environment. Another limitation of the study is the inclusion of professionals from a single municipality, which does not allow for the generalization of the results. However, this study distinguishes itself by allowing for the understanding of coping strategies adopted by PHC workers and promoting the development of interventions that can strengthen the negative aspects and consolidate the positive aspects of this reality.

## 5. Conclusions

This study showed that PHC professionals adopt different coping strategies for the problems and stressful situations experienced in the work environment. These strategies can involve more positive attitudes aimed at management and problem solving, and emotional responses that involve attitudes of avoidance and denial of the problem.

Considering that confronting problems in a positive manner is the best way to deal with the oppressive situations that arise in the work environment, it is essential that managers create support structures that encourage and strengthen the capacity of PHC professionals to adopt assertive strategies to face and solve problems.

## Figures and Tables

**Table 1 ijerph-19-03332-t001:** Sociodemographic and professional characteristics of primary healthcare (PHC) workers in São José do Rio Preto, São Paulo, Brazil.

Variables	n	%
**Professional Category**		
Physician	32	9.6
Nurse	20	6.0
Nursing Auxiliary/Technician	71	21.3
Community Health Agent	210	63.1
**Gender**		
Male	60	18.0
Female	270	81.1
No answer	3	0.9
**Age Group (years)**		
18 to 28	1	0.3
29 to 39	32	9.6
40 to 59	138	41.4
60 or more	149	44.7
No answer	13	3.9
**Education**		
High School	149	44.7
Higher Education/Graduate	184	55.3
**Civil Status**		
Married/Stable Union	210	63.1
Single	86	25.8
Separated	28	8.4
Widowed	9	2.7
**Contract Type**		
Government employee	248	74.5
Hired	80	24.1
No answer	3	0.9
**Weekly Work Hours**		
20 h	20	6.0
30 h	8	2.4
40 h	303	91.0
No answer	2	0.6
**Family Income (number of minimum wages *)**		
Up to one	23	6.9
Two to five	189	56.8
Six to 10	66	19.8
More than 10	47	14.1
No answer	8	2.4
**Length of time working in PHC**		
Up to two years	126	37.8
>two and ≤five years	56	16.8
>five and ≤10 years	64	19.2
Over 10 years	71	21.3
No answer	16	4.8
**Satisfied with Profession**		
Yes	259	77.8
No	70	21.0
No answer	4	1.2

* Minimum wage value: BRL 937.00 ≈ USD 299.35 (1 USD = 3.1301 BRL).

**Table 2 ijerph-19-03332-t002:** Mean scores of the coping strategies of primary healthcare workers in São José do Rio Preto, São Paulo, Brazil.

Coping Strategy	Mean Score	Standard Deviation	95% CI *	*p*-Value (*t*-Test)
Problem-focused	3.8	0.66	3.7–3.9	<0.001
Emotion-focused	2.4	0.62	2.4–2.5	
Religious Practices/Fantastical Thinking	3.2	0.78	3.1–3.3	
Pursuit of Social Support	3.1	0.91	3.0–3.2	

* 95% Confidence Interval.

**Table 3 ijerph-19-03332-t003:** Mean scores of the coping strategies of PHC workers, according to sociodemographic and professional variables. São José do Rio Preto, São Paulo, Brazil.

Coping Strategy	Problem-Focused	Emotion-Focused	Religious Practices/Fantastical Thinking	Pursuit of Social Support
	Mean (SD *)	Mean (SD *)	Mean (SD *)	Mean (SD *)
Professional Category				
Physician	3.9 (1.5)	3.6 (1.8)	2.3 (1.9)	3.0 (1.5)
Nurse	3.5 (1.4)	3.6 (1.3)	3.2 (1.4)	3.7 (1.4)
Nurse Auxiliary/Technician	3.7 (1.3)	4.0 (1.2)	3.7 (1.2)	3.7 (1.3)
Community Health Agent	3.8 (1.3)	4.3 (1.4)	4.2 (1.2)	3.8 (1.3)
*p*-value ****	0.136	0.091	*<0.001*	*0.045*
Gender				
Male	3.9 (0.7)	2.5 (0.7)	2.9 (0.8)	2.8 (1.0)
Female	3.8 (0.7)	2.4 (0.6)	3.3 (0.8)	3.2 (0.9)
*p*-value *****	0.495	0.872	*0.001*	*0.005*
Age Group (years)				
18 to 28	3.4 (0.8)	3.8 (0.6)	3.8 (0.5)	3.8 (0.5)
29 to 39	3.3 (0.5)	3.5 (0.7)	3.5 (0.7)	3.5 (0.7)
40 to 59	3.4 (0.8)	3.4 (0.7)	3.4 (0.7)	3.4 (0.7)
60 or more	3.4 (0.8)	2.8 (0.5)	3.3 (0.7)	3.4 (0.8)
*p*-value *****	0.926	*0.014*	0.235	0.271
Education				
High School	1.5 (0.5)	1.6 (0.5)	1.8 (0.4)	1.9 (0.4)
Higher Education/Graduate	1.6 (0.5)	1.6 (0.5)	1.4 (0.5)	1.6 (0.5)
*p*-value *****	0.860	0.111	*<0.001*	0.168
Family Income (number of minimum wages ^§^)				
Up to one	2.2 (0.4)	2.6 (1.2)	3.5 (1.0)	2.9 (1.0)
Two to five	2.5 (0.8)	2.5 (0.8)	2.6 (0.8)	2.5 (0.8)
Six to 10	2.4 (0.8)	2.4 (0.8)	2.5 (0.8)	2.4 (0.8)
More than 10	2.5 (0.8)	2.3 (0.5)	2.3 (0.8)	2.4 (0.8)
*p*-value ****	0.701	0.259	*0.001*	0.290
Length of time working in PHC				
Up to two years	3.3 (0.6)	2.4 (1.3)	2.5 (1.0)	3.3 (1.4)
> two and ≤ five years	2.5 (1.2)	2.6 (1.3)	2.6 (1.3)	2.4 (1.3)
> five and ≤ 10 years	2.2 (1.3)	2.2 (1.3)	2.6 (1.3)	2.3 (1.3)
Over 10 years	2.2. (1.4)	2.0 (1.2)	2.0 (1.3)	2.4 (1.3)
*p*-value ****	0.184	*0.008*	*<0.001*	0.795
Satisfied with Profession				
Yes	3.8 (0.7)	2.4 (0.6)	3.2 (0.8)	3.1 (0.9)
No	3.8 (0.7)	2.6 (0.6)	3.4 (0.7)	3.1 (0.9)
*p*-value *	0.921	*0.011*	*0.010*	0.685

* Standard Deviation. ** *t*-test. *** ANOVA test. **^§^** Minimum wage value: BRL 937.00 ≈ USD 299.35 (1 USD = 3.1301 BRL).

## Data Availability

The datasets generated during the current study are not publicly available but are available from the corresponding author on reasonable request.

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
