# Peer review of "Analysis of the Coping Strategies of Primary Health Care Professionals: Cross-Sectional Study in a Large Brazilian Municipality"

_ijerph, 2022, doi:10.3390/ijerph19063332_

Round 1
Reviewer 1 Report
After reviewing the article, I believe it is convenient to improve the theoretical part, that is, to include a greater wealth of citations and authors that improve the practical part of the article. Similarly, the sample section should be redrafted and further explain the characteristics of the sample. In the results section, I think it is more convenient to describe the results since the tables appear but the results are not defined.Author Response
We thank the reviewer for all his notes and recommendations. We have tried, based on the reviewer's opinion, to adjust and improve the manuscript. The changes we have made to the text to meet the recommendations of reviewer are yellow highlighted.
- After reviewing the article, I believe it is convenient to improve the theoretical part, that is, to include a greater wealth of citations and authors that improve the practical part of the article.
Response: We have revised the introduction and included new citations, to improve the practical part of the article.
- Similarly, the sample section should be redrafted and further explain the characteristics of the sample.
Response: We have reworded the sample section to meet this recommendation.
- In the results section, I think it is more convenient to describe the results since the tables appear but the results are not defined.
Response: We have revised the way the results are presented, including the score values described in the tables.

Reviewer 2 Report
- I ask the authors to write the abstract respecting its recommended structure.
- A sentence regarding the purpose and objectives of the study must be written because it exists in the abstract and is missing in the article.
- The methodology must be completed with a complete description of the proposed questionnaire, how to use it and how to interpret the results. The items are numbered, but I don't know what these numbers mean.
- I suggest to the authors to publish a study with the validation of the proposed questionnaire and then to publish the results of its application.
- The results presented are incomplete. Only mean scores on sections were presented in the tables and the score per item is missing.
- The results and p-value must be written in parentheses. (Only p-value has no scientific relevance if you do not know what the result is).
- In the tables, the word sex must be replaced by gender.
- There are too many bibliographic references to the same sentence. This suggests that they were forcibly placed in the text. I suggest that there be a maximum of 2 references to a sentence.
- The bibliography must be corrected and DOI must be checked for each reference. At reference 49, DOI is unavailable.
Author Response
We thank the reviewer for all his notes and recommendations. We have tried, based on the reviewer's opinion, to adjust and improve the manuscript. The changes we have made to the text to meet the recommendations of reviewer are blue highlighted.
- I ask the authors to write the abstract respecting its recommended structure.
Response: We structure the abstract as requested (Objective, Methods, Results, Conclusions).
- A sentence regarding the purpose and objectives of the study must be written because it exists in the abstract and is missing in the article.
Response: The purpose of the study is in the last sentence of the introduction. We have adjusted it according to what is presented in the abstract.
- The methodology must be completed with a complete description of the proposed questionnaire, how to use it and how to interpret the results. The items are numbered, but I don't know what these numbers mean.
Response: We include the description of the items that are grouped together to make up each of the four factors of the EMEP scale.
- I suggest to the authors to publish a study with the validation of the proposed questionnaire and then to publish the results of its application.
Response: As we mentioned in the Methods, the EMEP scale has been developed and validated in Brazil by Seidl, Tróccoli, and Zannon, from the version adapted to Portuguese.
- The results presented are incomplete. Only mean scores on sections were presented in the tables and the score per item is missing.
Response: As described in the Method, the EMEP scale consists of 45 items, which are grouped to evaluate four factors: factor 1 - problem-focused coping strategies; factor 2 - emotion-focused coping strategies; factor 3 - pursuit of religious practices/fantastical thinking; factor 4 - pursuit of social support. In presenting the results, we considered the analysis of the four factors that represent the different coping strategies adopted by the professionals.
- The results and p-value must be written in parentheses. (Only p-value has no scientific relevance if you do not know what the result is).
Response: We revised the way the results are presented, including the score values along with the p-value).
- In the tables, the word sex must be replaced by gender.
Response: We have done the sex to gender substitution as recommended.
- There are too many bibliographic references to the same sentence. This suggests that they were forcibly placed in the text. I suggest that there be a maximum of 2 references to a sentence.
Response: We have revised the text and kept to a maximum of two references for a sentence, as recommended.
- The bibliography must be corrected and DOI must be checked for each reference. At reference 49, DOI is unavailable.
Response: We have reviewed the references and checked all the DOI and access links.

Reviewer 3 Report
I have reviewed the manuscript entitled "Analysis of the coping strategies of primary health care professionals: cross-sectional study in a large Brazilian municipality". I think the research is interesting and it could be published in this journal. I suggest presenting in a more detailed way what are the emotions studied in the research, focusing on the importance to share these in the social interaction from a social psychology perspective. The same thing about the gender that I observe is an important topic in the results. I think it is important to incorporate some perspective and cite authors and research published in this journal.
The last thing is to offer more context about the data collected. It is important to understand the Brazilian health care context.
These are some papers that can be revised to improve the manuscript:
Belli, S. (2018). Managing Negative Emotions in Online Collaborative Learning A multimodal approach to solving technical difficulties. Digithum, (22).
Gengler, A. M. (2020). Emotions and medical decision-making. Social Psychology Quarterly, 83(2), 174-194.
Villanueva-Felez, A., Woolley, R., & Cañibano, C. (2015). Nanotechnology researchers’ collaboration relationships: A gender analysis of access to scientific information. Social studies of science, 45(1), 100-129.
Author Response
We thank the reviewer for all his notes and recommendations. We have tried, based on the reviewer's opinion, to adjust and improve the manuscript. The changes we have made to the text to meet the recommendations of reviewer are green highlighted.
- I have reviewed the manuscript entitled "Analysis of the coping strategies of primary health care professionals: cross-sectional study in a large Brazilian municipality". I think the research is interesting and it could be published in this journal. I suggest presenting in a more detailed way what are the emotions studied in the research, focusing on the importance to share these in the social interaction from a social psychology perspective.
Response: We reviewed the text of the introduction and we emphasize that "we understand that PHC professionals may adopt different coping strategies when facing stressful situations in the work environment. In this study, we intend to explore which coping strategies PHC professionals use to adapt to stress. We understand that knowing the different coping strategies adopted by these profession-als can contribute to the implementation of policies that promote positive coping mechanisms and improve the quality of life and productivity of this population, and by realizing this, we can improve the mental health of PHC workers".
The same thing about the gender that I observe is an important topic in the results.
Response: We did the discussion of the results considering the gender results obtained.
I think it is important to incorporate some perspective and cite authors and research published in this journal.
Response: We have cited references from this journal in our manuscript.
[22]. Puto, G.; Jurzec, M.; Leja-Szpak, A.; Bonior, J.; Muszalik, M.; Gniadek, A. Stress and Coping Strategies of Nurses Working with Patients Infected with and Not Infected with SARS-CoV-2 Virus. Int. J. Environ. Res. Public Health 2022, 19, 195. https://doi.org/10.3390/ijerph19010195
[39]. Marcisz-Dyla, E.; Dąbek, J.; Irzyniec, T.; Marcisz, C. Personality Traits, Strategies of Coping with Stress and Psychophysical Wellbeing of Surgical and Non-Surgical Doctors in Poland. Int. J. Environ. Res. Public Health 2022, 19, 1646. https://doi.org/10.3390/ijerph19031646
- The last thing is to offer more context about the data collected. It is important to understand the Brazilian health care context.
Response: We have reviewed the text to better contextualize the Brazilian health system and the Primary Health Care level, where the professionals in this study work.
These are some papers that can be revised to improve the manuscript:
Belli, S. (2018). Managing Negative Emotions in Online Collaborative Learning A multimodal approach to solving technical difficulties. Digithum, (22).
Gengler, A. M. (2020). Emotions and medical decision-making. Social Psychology Quarterly, 83(2), 174-194.
Villanueva-Felez, A., Woolley, R., & Cañibano, C. (2015). Nanotechnology researchers’ collaboration relationships: A gender analysis of access to scientific information. Social studies of science, 45(1), 100-129.
Response: We thank you for indicating the texts. They were important for the revision process of this manuscript.

Round 2
Reviewer 1 Report
The authors have made a correct modification of the article in response to the proposed comments.Author Response
Dear reviewer,
We thank you for your important contributions to the improvement of our manuscript.
The authors.
Reviewer 2 Report
Thanks to the authors for their work.
I agree with the improvements made.
Please write p-value correctly in the text (p written in italics).
Author Response
Dear reviewer,
We thank you for your important contributions to the improvement of our manuscript.
We have made the adjustments to the p-value (p written in italics) in the text as requested.
Best regards,
The authors.
